

# Historical connections among river basins and climatic changes explain the biogeographic history of a water rat

Jeronymo Dalapicolla[1,2] and Yuri Luiz Reis Leite[1]

[1] Departamento de Ciências Biológicas, Universidade Federal do Espírito Santo, Vitória, Espírito Santo, Brazil
[2] Departamento de Ciências Biológicas, Escola Superior de Agricultura "Luiz de Queiroz", Universidade de São Paulo, Piracicaba, São Paulo, Brazil

## ABSTRACT

**Background**. The water rat *Nectomys squamipes* (Cricetidae: Sigmodontinae) is a semiaquatic rodent from eastern South America that shows shallow genetic structure across space, according to some studies. We tested the influence of hydrography and climatic changes on the genetic and phylogeographic structure of this semiaquatic small mammal.

**Methods**. DNA sequences of two mitochondrial genetic markers (Cyt b and D-loop) and six microsatellite loci from water rats were collected at 50 localities in five river basins in the Atlantic Forest along the eastern coast of South America. We evaluated the genetic structure within and among river basins, and we estimated divergence dates. Species distribution models for the present and past were built to identify possible gene flow paths.

**Results**. Mitochondrial data and species distribution models showed coherent results. Microsatellite loci showed a more complex pattern of genetic differentiation. The diversification of *N. squamipes* haplotypes occurred during the Pleistocene and the river basin cannot explain most of the genetic structure. We found evidence of population expansion during the last glacial maximum, and gene flow paths indicate historical connections among rivers in the Atlantic Forest.

**Discussion**. Historical connections among rivers in the Atlantic Forest may have allowed *N. squamipes* to disperse farther across and within basins, leading to shallow genetic structure. Population expansions and gene flow through the emerged continental shelf during glacial period support the Atlantis forest hypothesis, thus challenging the forest refuge hypothesis.

## INTRODUCTION

In recent years, many phylogeographic studies have tested different hypotheses to explain the diversity and genetic structuring patterns found in vertebrates in the Atlantic Forest, eastern South America (*Thomé et al., 2014*). Quaternary climatic cycles have been considered the main drivers of population structure and genetic diversity but two opposing views regarding glacial periods emerged recently: population bottlenecks and stable

Corresponding author
Jeronymo Dalapicolla,
jdalapicolla@gmail.com

refuges characterize the forest refuge hypothesis (FRH) (*Carnaval & Moritz, 2008*), while population expansion onto the continental shelf is the main argument behind the Atlantis forest hypothesis (AFH) (*Leite et al., 2016*). The FRH is based upon forest retraction and fragmentation during glacial periods that would have caused isolation and consequently allopatric speciation within forest patches, or refuges, surrounded by open habitats (*Haffer, 1969*; *Vanzolini & Williams, 1970*). Although originally formulated to account for species diversity in the Amazon, it was later adapted to explain high contemporary diversity and endemism at historically stable Atlantic Forest areas, or refuges (*Carnaval & Moritz, 2008*; *Valdez & D'Elía, 2013*). In this model, population size reductions are expected in unstable areas due to habitat loss during glacial periods and population stability or expansion during interglacials (*Leite & Rogers, 2013*). The AFH, on the other hand, claims that forest specialist species actually expanded during the last glacial period (Last Glacial Maximum, LGM), following the expansion of the Atlantic Forest onto the emerged Brazilian continental shelf (*Leite et al., 2016*). Similarly to South-East Asia, where widespread rainforest covered the exposed Sunda shelf during the LGM (*Wang et al., 2009*; *Raes et al., 2014*), the marine regressions on the Brazilian coast exposed land where large stretches of rainforest would have flourished, leading to population expansion of forest species during glacial periods (*Leite et al., 2016*).

The present study deals with the phylogeographic structure of a water rat, *Nectomys squamipes* (Brants), in the Atlantic Forest of eastern Brazil. The genetic structure of semiaquatic mammals may have been shaped differently from terrestrial mammals because the displacement through rivers may allow semiaquatic mammals to disperse further, leading to less genetic differentiation among populations (*Centeno-Cuadros et al., 2011*). Consequently, the phylogeographic patterns would be more similar to those of large-sized mammals with great dispersal ability with shallow genetic structure. Alternatively, the specialisation to their habitat may constrain their dispersal patterns and, subsequently, the gene flow among populations, leading to a genetic phylogeographic structure that is more distinct that of other land mammals (*Laurence, Smith & Schulte-Hostedde, 2013*).

In the Atlantic Forest, *N. squamipes* is adapted to semiaquatic life, living along river courses and having a broad geographic distribution, ranging from areas of Atlantic Forest to Cerrado (*Bonvicino & Weksler, 2015*). This species has a small home range (0.7–3.96 ha), and the largest distance recorded between site of capture and a river course was approximately 520 m (*Pires et al., 2002*). Its displacements occur along riverbanks, and they are seldom farther than 7 m away from the margins (*Lima, Pinho & Fernandez, 2016*). Therefore, high genetic divergence is expected among *N. squamipes* populations from different basins while individuals within the same basin should have low genetic divergence.

A study on *N. squamipes* nucleotide variation found a relevant genetic structure (*Maroja, Almeida & Seuánez, 2003*), while two others found a shallow structuring among populations of a broader geographical area (*Almeida et al., 2000a*; *Almeida et al., 2005*). Using randomly amplified polymorphic DNA (RAPD) (*Almeida et al., 2000a*) and microsatellites (*Almeida et al., 2005*), these authors concluded that the degree of genetic differentiation was low, contradicting the ecological data about the dispersal ability of *N. squamipes*.
Here we integrated analyses of mitochondrial and nuclear molecular markers with species distributions modelling to comprehend the phylogeographic structure of *N. squamipes*. Our goals were to: (1) evaluate evolutionary hypotheses to explain the biogeographic history of the water rat; (2) test if the genetic structure is strongly influenced by hydrography, leading to low gene flow and high divergence among basins, or if rivers allow *N. squamipes* to disperse across basins, leading to homogeneity and low genetic structure.

## MATERIALS & METHODS

### Molecular data

We used 161 samples from 50 localities distributed throughout the range of *N. squamipes* (Fig. 1). Two DNA sequence was obtained from GenBank (http://www.ncbi.nlm.nih.gov/genbank/), five were donated by researchers, and 154 liver and muscle samples preserved in ethanol were supplied by three mammal collections: CTA-UFES (Coleção de Tecidos Animais da Universidade Federal do Espírito Santo, Vitória, Brazil), MBML (Museu de Biologia Professor Mello Leitão, Santa Teresa, Brazil), and MCN-M (Museu de Ciências Naturais PUC Minas, Belo Horizonte, Brazil) (see Appendix S1 in Supporting Information). We sequenced 801 bp of cytochrome b (Cyt b) and 425 bp from the control region of mitochondrial DNA (D-loop) (see Appendix S2, Table S2.1). The PCR products were purified with the enzymes ExoSAP (GE Healthcare Life Sciences, Little Chalfont, UK) and sequenced using Big Dye 3.1 in the sequencer ABI 3500 (Applied Biosystems, Life Technologies Corporation, Waltham, MA, USA). We carefully examined the electropherograms for both markers to check for noise and double peaks. We also had the Cyt b sequences aligned and translated to check for stop codons. Any sequences showing strong noise, double peaks or stop codons (for Cyt b) were discarded.

Sequences were used to obtain phylogenies through maximum parsimony (MP) in MEGA 5.2 (*Tamura et al., 2011*), Bayesian inference (BI) in MrBayes 3.2.2 (*Ronquist et al., 2012*), maximum likelihood (ML) in Garli 2.01 (https://code.google.com/archive/p/garli/) and in PhyML 3.0 (*Guindon & Gascuel, 2003*). The model of nucleotide substitution was selected using jModelTest 2.1.4 (*Darriba et al., 2012*) for Cyt b and D-loop individually. For the concatenated analysis, the data were partitioned, and the specific model was applied to each marker. Clade support was estimated through 100 bootstrap replications for MP and ML. The BI analysis was performed for $1.2 \times 10^6$ generations for Cyt b, $3 \times 10^6$ generations for D-loop, and $2.5 \times 10^7$ generations for the concatenated data, always with 2 runs with trees sampled every 1,000 generations. We confirmed that effective sample sizes (ESSs) were above 200 in Tracer 1.6 (*Rambaut et al., 2014*) and then discarded the first 25% as burn-in. *Nectomys rattus* Pelzeln, *Nectomys apicalis* Peters, *Amphinectomys savamis* Malygin, *Holochilus chacarius* Thomas, *Pseudoryzomys simplex* (Winge), *Oligoryzomys flavescens* (Waterhouse), *Oligoryzomys chacoensis* (Myers & Carleton) and *Oligoryzomys nigripes* (Olfers) were used as outgroups.

Beast 1.8.2 (*Drummond & Rambaut, 2007*) was used to date divergences among clades of Cyt b haplotypes, through 100 million generations using the lognormal relaxed clock and Yule process. Four secondary calibration points were defined based on studies that

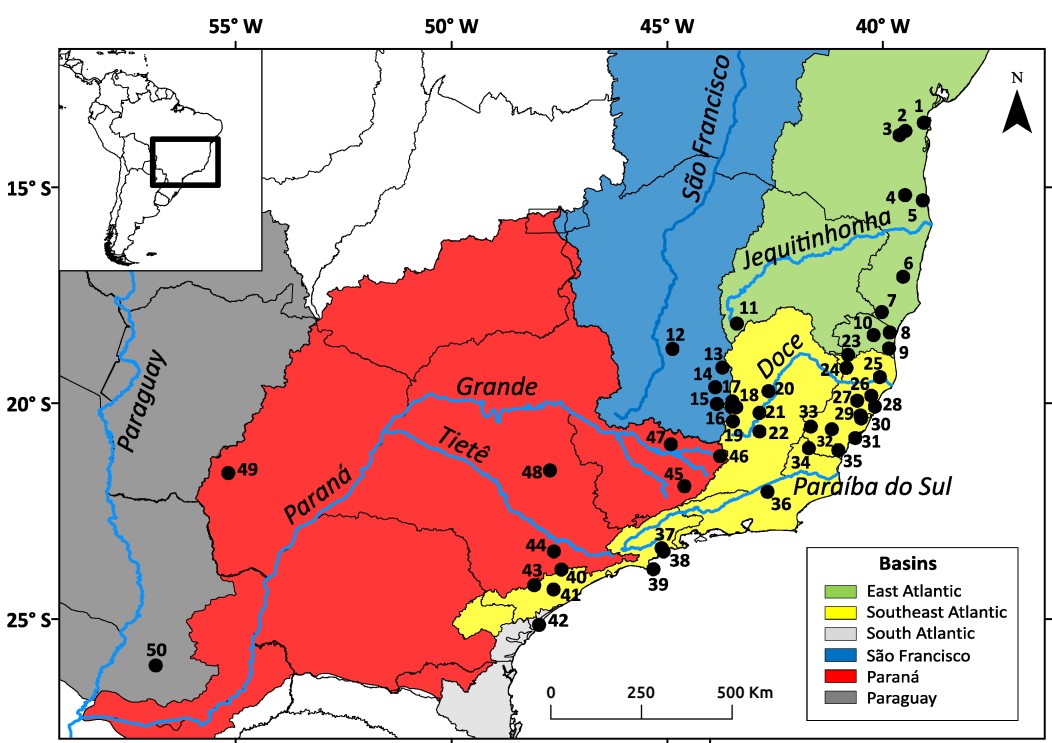

**Figure 1** **Geographic distribution of the *Nectomys squamipes* samples.** The colors represent the basins with main rivers. Numbers correspond to localities in Appendix S1.

used multiple fossils for estimating clades ages within Sigmodontinae. The divergence between *Pseudoryzomys* and *Holochilus* at 2.58 million years ago (Ma) (0.43–5.38 Ma), and the divergence between *Cerradomys* and *Sooretamys* at 2.35 Ma (0.42–5.02 Ma), are from *Machado et al. (2014)*, and the divergence among *O. nigripes*, *O. flavescens*, and *O. chacoensis* at 3.71 ± 0.035 Ma (*Palma et al., 2010*), and the divergence between *Amphinectomys* and *Nectomys* at 2.64 Ma (2.00–3.37 Ma) are from *Leite, Kok & Weksler (2015)*. The secondary calibrations priors were normally distributed, using the 95% confidence interval, following recommendations in *Hipsley & Müller (2014)*. A population dynamics plot (Bayesian skyline plot) was created using Cyt b sequences in Beast 1.8.2, covering 20 million generations and using the lognormal relaxed clock with the mutation rate found while dating the clades.

We grouped samples according to the river basins. Hydrographic boundaries followed the classification set by the Brazilian National Water Agency, using HidroWeb (http://www.snirh.gov.br/hidroweb/) and GeoNetwork (http://metadados.ana.gov.br/geonetwork/srv/pt/main.home). We performed Bayesian Analysis of Population Structure (BAPS 6.0) (*Corander & Marttinen, 2006*) to estimate the number of genetic clusters present in each mitochondrial markers separately, and we compared them with our clusters based on river basin boundaries. We used a mixture model, clustering of linked molecular data, and tested $K = 1$–10 with 20 replicates per $K$. Haplotype networks were constructed

in Network 4.6 (*Bandelt, Forster & Röhl, 1999*) using the median-joining tool and only sequences without ambiguities. Tests for neutrality (Tajima's D; $R_2$; Fu's Fs), and genetic divergence ($F_{ST}$) (*Weir & Cockerham, 1984*) among clades, BAPS clusters, and river basins were performed in DnaSP 5.10 (*Librado & Rozas, 2009*), and Arlequin 3.5 (*Excoffier & Lischer, 2010*). The genetic distances within and among clades, BAPS clusters, and river basins were calculated in MEGA 5.2 (*Tamura et al., 2011*) using the *p*-distance.

We used six microsatellite loci as nuclear markers: Nec12, Nec14, Nec15, Nec18, (*Almeida et al., 2000b*), Nec19, and Nec23 (*Maroja et al., 2003*) (Table S2.1). The automated sequencer ABI 3500 was used for electrophoresis with size standard GeneScan 600 LIZ® and allelic calling using GeneMapper 4.1 (Applied Biosystems; Life Technologies Corporation, Waltham, USA). Allele sizes were corrected to avoid allelic drift using TANDEM 1.09 (*Matschiner & Salzburger, 2009*). Our sample was divided into five groups representing the river basins. Basic descriptive statistics, such as number of alleles, expected and observed heterozygosity, gene diversity, and allelic richness were calculated for each locus and for each river basin using GENEPOP 4.2 (*Rousset, 2008*), and FSTAT 2.9.3.2 (*Goudet, 2001*). GENEPOP 4.2 was also used to test the linkage disequilibrium, null alelles frequency, and for deviation from Hardy-Weinberg equilibrium (HWE) based on Fisher's method. $R_{ST}$ values were calculated in Arlequin 3.5 (*Excoffier & Lischer, 2010*) using 10,000 replications.

BOTTLENECK 1.2.02 (*Piry, Luikart & Cornuet, 1999*) was used to infer recent expansions or bottlenecks within river basins using Two-Phased Model (TPM) and Stepwise Mutation Model (SSM). STRUCTURE 2.3.4 (*Pritchard, Stephens & Donnelly, 2000*) was used to identify the genetic structure, with *a priori* classification by river basins using Locprior in admixture model, as suggested by *Hubisz et al. (2009)* and *Pritchard, Wen & Falush (2010)*, where the amount of available data is very limited (few markers or few individuals) or when the models do not provide indication of genetic structure. We used 100,000 burn-in steps followed by 300,000 Markov chain Monte Carlo (MCMC) simulation steps with 20 independent runs for each value of $K$ (1–6). Structure Harvester (*Earl & Von Holdt, 2012*) was used for choosing the best number of assumed clusters ($K$) based on the second order rate of change of the likelihood ($\Delta K$) (*Evanno, Regnaut & Goudet, 2005*), and for evaluating the standard deviation of simulations for each $K$. CLUMPAK (*Kopelman et al., 2015*) was also used to produce the bar graphs for the best $K$.

For all genetic markers we performed analysis of molecular variance (AMOVA), grouping our samples according to the river basins, and to localities inside the basins. Localities with only one sample were grouped with the nearest locality within the same river basin. We used only sequences without ambiguities, and genotyped samples with one missing locus at most. We conducted AMOVA in Arlequin 3.5 (*Excoffier & Lischer, 2010*), using $F_{ST}$ (mtDNA), and $R_{ST}$ (microsatellites) with 10,000 replications.

## Species distribution modelling (SDM)

We used MaxEnt 3.3.3k (*Phillips, Anderson & Schapire, 2006*) and models were built using 30% of the points for testing and the analysis was performed with the average model of 30 replications constructed through the bootstrap. The geographical area contemplated the five

river basins in Brazil where the species is distributed: Paraná, São Francisco, South Atlantic (including Ribeira do Iguape river), Southeast Atlantic (including Doce and Paraíba do Sul rivers), and East Atlantic (including Jequitinhonha river). Since one of our goals was to infer palaeomodels, we used only 19 climate variables from Worldclim (*Hijmans et al., 2005*), which correspond to the variables available for the past. We modelled the present (1950–2000), the Last Glacial Maximum (LGM) ca. 22 ka, and the Last Interglacial (LIG) ca. 120–140 ka, all in 2.5 min resolution (approximately 5 km$^2$). For the LGM, we used the MIROC-ESM model (*Watanabe et al., 2011*) and for the LIG we used the model by *Otto-Bliesner et al. (2006)*.

A database with 463 georeferenced locations was compiled based on data from museums (242 points), scientific articles (95), theses and dissertations (126). The museum data were obtained from SpeciesLink (http://splink.cria.org.br/), Global Biodiversity Information Facility (GBIF: http://www.gbif.org), and Arctos Collaborative Collection Management Solution (http://arctos.database.museum/), or requested from curators. The points of occurrence were reviewed, and only data with precise specific locations were used. The locations of the points of occurrence were verified using Google Maps (http://maps.google.com.br), and two tools from SpeciesLink (http://splink.cria.org.br/) were used when compiling our locality database: (1) "geoLoc", which searches for place names in gazetteers, was used to georeference localities with no coordinates; (2) "infoXY", was used to fill out administrative data (country, state, county) for a given lat-long coordinate. The coordinates found in "geoLoc" were not used when building the models, only in the external validation test.

We selected only the most important variables indicated by the Principal Component Analysis (PCA) and those with correlation values less than 0.7 among them. Theses tests were perfomed in R 3.0.1 (R Foundation for Statistical Computing, Vienna, Austria) using "raster" (*Hijmans & Van Etten, 2015*) and "vegan" packages (*Oksanen et al., 2015*). Moreover, to reduce the spatial autocorrelation due to the sample bias, we used rarefy by environmental heterogeneity method using a 10 km buffer in SDMtoolbox 1.1c (*Brown, 2014*) in ArcGIS 10.1 (Environmental Systems Research Institute, Redlands, CA, USA). Rarefy by environmental heterogeneity method was also used to reduce the spatial autocorrelation of the points in the external validation. This resulted in 109 occurrence points: 69 were used to create the SDM and 40 for external validation (see Table S2.2).

To estimate optimal model complexity for the SDM, we used "ENMeval" package in R 3.0.1 (*Muscarella et al., 2014*). We tested eight different feature class (FC) combinations (L, LQ, LQP, H, T, LQH, LQHP, LQHPT; where $L$ = linear, $Q$ = quadratic, $H$ = hinge, $P$ = product and $T$ = threshold); with regularization values (RM) from 0.5 to 3.0 (increments of 0.5), and we used block method for partitioning occurrence data, which resulted in 48 individual model runs. The best model was one with minimum Akaike information criterion value (*i.e.* $\Delta$AICc $= 0$), FC = LQH and RM = 2.0. We evaluated this model with indices in MaxEnt validation and external validation step. We analyzed the area under the curve (test AUC), default values and the *p*-value for the three thresholds frequently used in MaxEnt: minimum training presence logistic threshold, 10 percentile training presence

logistic threshold, and maximum test sensitivity plus specificity logistic threshold. Indices analyzed in the external validation were based on the 10% threshold.

We built scatter plots for the three environmental variables that contributed most to the SDM. The models were based on the 10% threshold and the environmentally appropriate area was calculated for each scenario (present, LGM and LIG) to evaluate whether there was variation in the size of the area over time.

### Landscape genetics

The lists of haplotypes or alleles shared among localities, and LGM and present-day SDMs were used to create gene flow paths among sampling locations, indicating possible migration routes for *N. squamipes*. We modelled gene pathways under LGM because recovered gene flow may reflect older demographic connections. The most recent Cyt b population expansion may have taken place during the LGM as suggested by the Bayesian skyline plot (see 'Results' below). The SDM was converted into a dispersal cost layer by inverting the model's suitability values (*Chan, Brown & Yoder, 2011*). Localities sharing haplotypes and alleles were connected by low cost paths using the dispersal cost map as a friction layer, in accordance with the method of *Chan, Brown & Yoder (2011)*. The software ArcGIS 10.1, and the package "arcpy" for Python v2.7.2 (Python Software Foundation, Delaware, USA) was used for these analyses.

## RESULTS

### Molecular data

Of the 161 initial samples, 83 Cyt b sequences and 149 D-loop sequences of *Nectomys squamipes* were obtained (77 and 148 sequences with no ambiguities, respectively); sequences for both markers were obtained from 70 samples (Table S2.3). The difference in the number of sequences obtained for the two genetic markers was largely due to double peaks and stop codons in the Cyt b sequences, possibly pseudogenes, which had been found in previous unpublished studies on *N. squamipes* (CR Bonvicino, National Cancer Institute of Brazil, pers. comm., 2014). Five D-loop sequences were discarded due to double peaks. For microsatellites, 152 individuals were genotyped, representing 43 localities, 120 samples with one missing locus at most (Table S2.4). None of six loci had deviation from HWE in global tests, the average frequency of null alleles per locus was always below 0.14 (Table S2.5), and no linkage disequilibrium was observed for any pair of loci (Table S2.6).

The best evolutionary model for Cyt b sequences was HKY + I + G and for D-loop was HKY + G. The trees with the best resolution and support (>70% for MP and ML, and >95% for BI) were those from Cyt b sequences, recovering three main clades: North, Central and South, in reference to their latitudinal ranges, regardless of the optimality criteria used (Fig. 2). The Central clade was not recovered in D-loop trees and in the BI tree with concatenated data; statistical support for these main clades were lower in D-loop and concatenated trees than Cyt b trees (see Appendix S3, Fig. S3.1 for D-loop and concatenated trees). The differences in topologies are due to the absence of D-loop sequences of *Nectomys apicalis* and *Amphinectomys savamis*. When we removed these outgroup sequences from the Cyt b matrix, we recovered a Cyt b topology similar to D-loop and concatenated trees

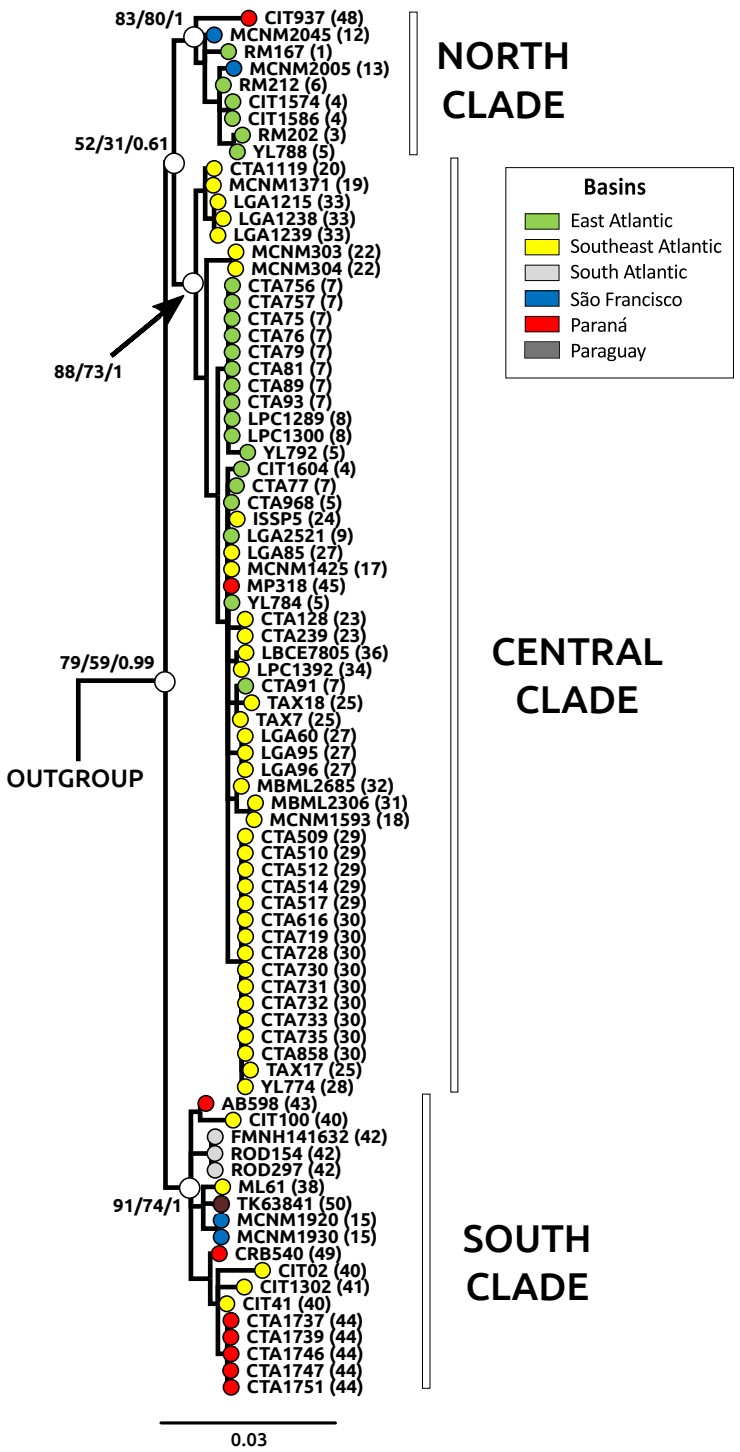

**Figure 2** **Phylogeny through Bayesian inference (BI) for Cyt b samples.** Main clades for *Nectomys squamipes* are represented with white circles, and their bootstrap support for maximum parsimony (MP), maximum likelihood (ML), and the posterior probability for Bayesian inference (BI) are also shown in the following order: MP/ML/IB. The topology of those main clades did not change in the different phylogenetics methods, only the statistical support. We chose the BI tree because it presented higher values of statistical support. Numbers correspond to localities in Appendix S1 and Fig. 1. The circle colors indicate the river basin where samples occur.

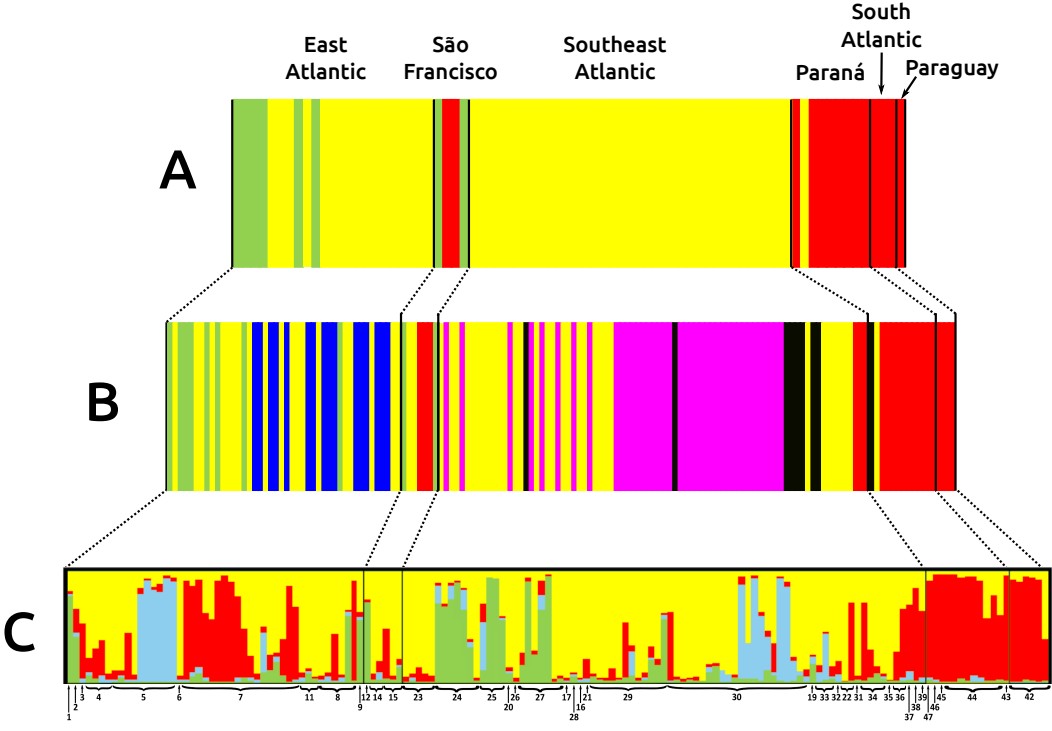

**Figure 3** Bar graphs for genetic structure in *Nectomys squamipes* using Cyt b (A), and D-loop (B) sequences in BAPS, and microsatellite data (C) in STRUCTURE. Each bar represents an individual organized by river basin and by latitude inside each basin. Black lines represent the basins limits, and each different colour a cluster, three for Cyt b, six for D-loop, and four for microsatellite data. Yellow represents the clusters related to Central clade in all markers, green to North clade, and red to South clade. Other colours were chose to improving the contrast. Numbers below bars correspond to localities in Appendix S1 and Fig. 1.

(Fig. S3.2). Samples from the coast of São Paulo, which belong to the Southeast Atlantic basin, were phylogenetically associated with samples from Paraná and South Atlantic basins, indicating a phylogeographic break in mitochondrial DNA at Serra da Bocaina, a mountain range between the states of São Paulo and Rio de Janeiro. The three main clades do not have a clear association with river basins. In addition, haplotype networks of mitochondrial DNA (Cyt b and D-loop, Fig. S3.3) showed three groups, associated with the main clades from the phylogenetic trees (North, Central, and South) but not with the river basins.

The three main clades of *N. squamipes* were also recovered in BAPS cluster analyses (Fig. 3). Cyt b data indicated three clusters, with the same composition of the three clades (Fig. 3A). D-loop recovered six clusters, one corresponding to the North clade, one to the South clade and four to the Central clade: Caparaó, North Doce River, South Doce River, and Southeast Atlantic cluster (Fig. 3B).

Microsatellite data recovered four genetic clusters ($K = 4$) for *N. squamipes* that are not related to the river basins (Fig. 3C). These clusters were similar to the Cyt b clusters (Fig. 3A)

and clades (Fig. 2), and are distributed across all basins, with different proportions depending on the locality (Fig. 3C). Microsatellite samples from southern Brazil (Paraná, and South Atlantic) are homogeneous, forming one group, whereas those from the other three basins (East Atlantic, São Francisco, Southeast Atlantic) are heterogeneous, showing different probabilities of belonging to three distinct groups, which are unrelated to the basins (Fig. 3C), but somewhat compatible with mitochondrial DNA clades (Fig. 2; Fig. S3.1).

Haplotype networks of mitochondrial DNA showed few mutational steps among groups (Cyt b and D-loop, Fig. S3.3), indicating low genetic divergence within *N. squamipes*. Genetic distances also showed similar results, small divergence within and among the *N. squamipes* clades, BAPS clusters, and river basins for Cyt b, and higher values for D-loop (Table S2.7). *Nectomys squamipes* has low intraspecific divergence (1.3% Cyt b; 2.7% D-loop), and the three intraspecific clades also display small genetic distances, with little intraclade (0.5–0.8% Cyt b; 1.5–1.7% D-loop) and interclade divergences for Cyt b (1.8–1.9%); interclade divergences for D-loop were higher (4.0–4.9%). In addition, samples from the São Francisco basin have greater intraregional diversity (1.6% Cyt b; 3.9% D-loop) than those from the other basins (0.1–1.2% Cyt b; 0.5–2.2% D-loop). The most recent common ancestor of all Cyt b haplotypes from *Nectomys* spp. dates back to the Plio-Pleistocene period 2.7–1.3 Ma, and Cyt b haplotypes from *N. squamipes* diverged 1.64–0.51 Ma (Fig. 4). The three clades found in *N. squamipes* separated 1,160,000–198,000 years ago in the Pleistocene (see Table S2.8). The mutation rate was estimated at 0.028 mutation/site/million years.

No significant values were found for most neutrality tests (Table S2.9). Samples from the Southeast Atlantic basin and the Central clade showed significant values in both markers. Significant values were also found for the Central (Cyt b) and Southeast Atlantic (D-loop) clusters in BAPS. These significant values indicate population expansions in the central area of the *N. squamipes* distribution. However, the effective population size estimated using microsatellite data indicated stability in the Southeast Atlantic basin and for *N. squamipes* as a whole (Table S2.10). These data also indicated population expansion in the Paraná basin and a recent bottleneck in the East Atlantic basin. Sample sizes were too small for estimating demographic histories of the São Francisco and the South Atlantic basins. Bayesian skyline plot showed continuous growth between the LIG and the LGM (120,000–20,000 years ago) for *N. squamipes*, and a decline toward the present, despite a wide confidence interval (Fig. 5).

We found different levels of population differentiation among river basins depending on the marker, and the river basin analysed (Table S2.11). The highest $F_{ST}/R_{ST}$ values were found in comparisons including the South Atlantic basin in all markers (Cyt $b =$ 0.36–0.56; D-loop = 0.19–0.39; Microsatellite = 0.09–0.13). Other river basins presented moderate values for $F_{ST}/R_{ST}$ (Cyt $b =$ 0.16–0.28; D-loop = 0.05–0.20; Microsatellite = 0.01–0.05). Among the genetic markers, Cyt b showed the highest values of population differentiation ($F_{ST} =$ 0.16–0.56), while the microsatellite showed the lowest values ($R_{ST} =$ 0.01–0.13). River basins explain only 11.08% of the genetic variation found in Cyt b, and were irrelevant for D-loop and microsatellite loci (Table 1). In addition, most of the genetic variation is within localities.

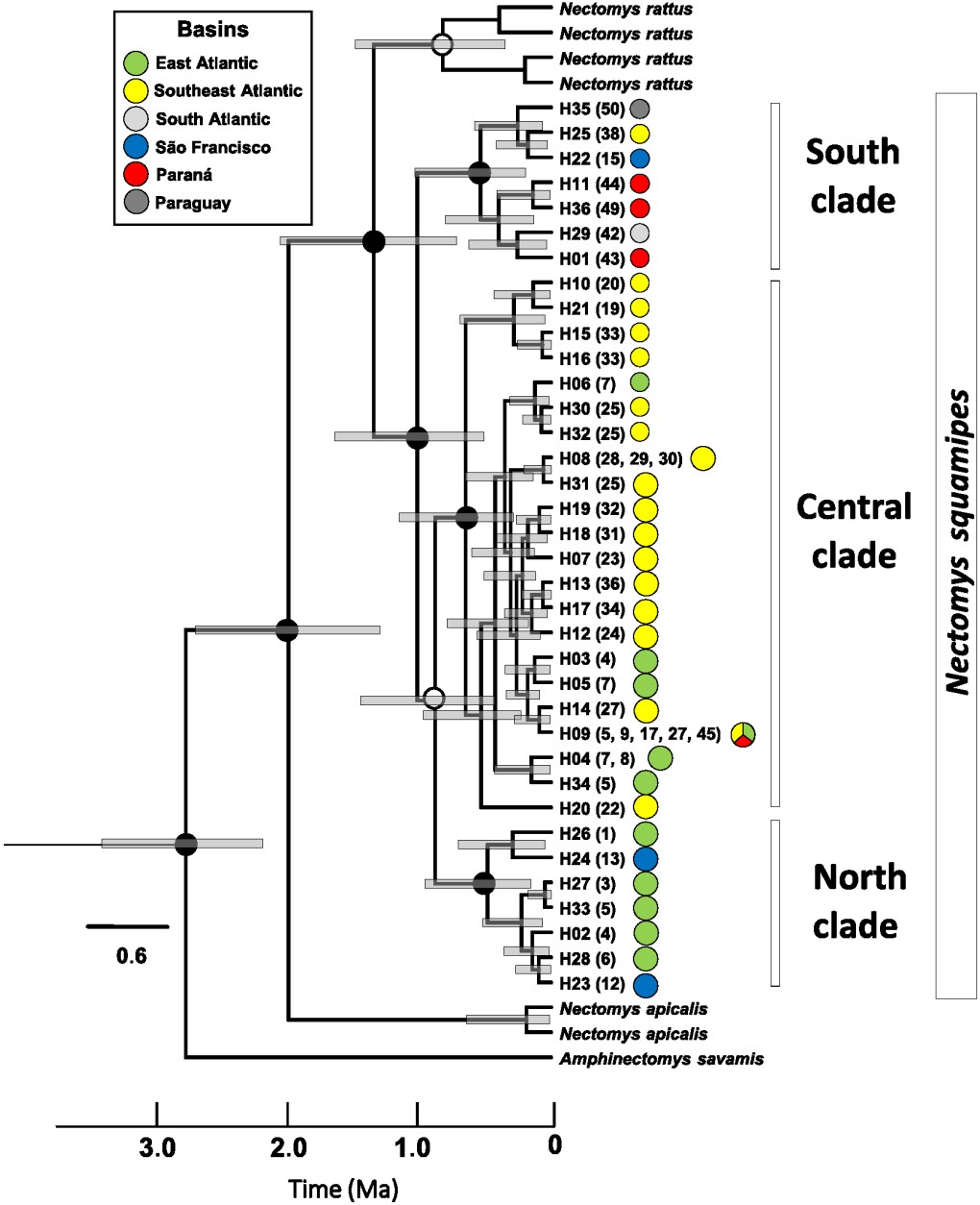

**Figure 4  Relaxed clock tree based on the Cyt b sequences of *Nectomys*.** Bars represent the confidence interval (95%). The exact mean values and confidence intervals are in Table S2.8. Localities numbers in parentheses are the same that Fig. 1, and they refer to Appendix S1 , and the haplotype list are in the Table S2.3. The circle colours indicate the river basin where haplotypes occur. Ma, millions years ago.

## Species distribution modelling

The SDM showed good statistical indices in the validation step (Table S2.12). The mean diurnal range of temperature (Bio 02) was the variable that contributed most to the model (62.02%), followed by the precipitation of warmest quarter (Bio 18: 20.39%), and the precipitation of wettest month (Bio 13: 14.42%) (Fig. S3.4).

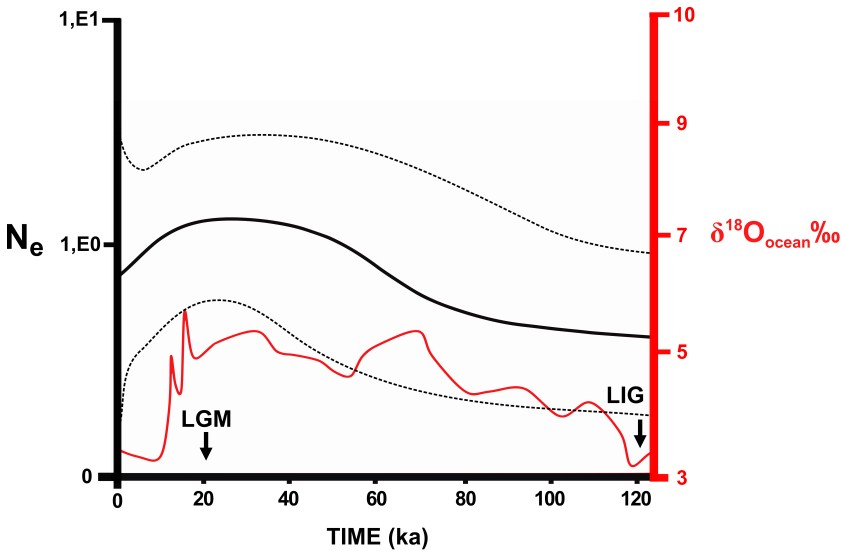

**Figure 5 Skyline plot showing the effective population size ($N_e$) of *Nectomys squamipes*.** The continuous black line is the effective population size × Generation Time ($N_e$) based on Cyt b sequences (with the 95% confidence interval in pointed line). We compared to the level of the stable isotope $\delta^{18}O_{ocean}$ ‰ (*Cohen, 2012*), corresponding to temperature oscillation (red line). The arrows indicate the period of the last glacial maximum (LGM) and the last interglacial (LIG). ka, thousands of years ago.

**Table 1 Analysis of molecular variance (AMOVA) for *Nectomys squamipes* considering river basins, localities, and the genetic markers.** Percentage of genetic variation, fixation indices and *p*-values attributable to heterogeneity among river basins, among localities within same basin, among localities and among individuals are presented in each line.

| Markers | Source of variation | d.f. | Sum of squares | Variance components | Variation (%) | Fixation indices | p-value |
|---------|--------------------|------|----------------|--------------------|--------------|--------------------|---------|
| Cyt b | Among basins | 4 | 7.13 | 0.05 | 11.08 | $F_{CT}$: 0.11 | 0.03 |
| | Among localities within basins | 15 | 14.97 | 0.21 | 42.54 | $F_{SC}$: 0.48 | <0.01 |
| | Within localities | 56 | 12.96 | 0.23 | 46.38 | $F_{ST}$: 0.54 | <0.01 |
| | Total | 75 | 35.05 | 0.5 | | | |
| D-loop | Among basins | 4 | 6.47 | <0.01 | 1.19 | $F_{CT}$: 0.01 | 0.32 |
| | Among localities within basins | 22 | 26.88 | 0.18 | 35.55 | $F_{SC}$: 0.36 | <0.01 |
| | Within localities | 121 | 38.29 | 0.32 | 63.26 | $F_{ST}$: 0.37 | <0.01 |
| | Total | 147 | 71.65 | 0.5 | | | |
| MS (all loci) | Among basins | 4 | 2,665.46 | 2.62 | 1 | $F_{CT}$: 0.01 | 0.24 |
| | Among localities within basins | 21 | 10,181.53 | 30.09 | 11.49 | $F_{SC}$: 0.12 | 0.02 |
| | Within localities | 212 | 48,574.63 | 229.13 | 87.5 | $F_{ST}$: 0.12 | <0.01 |
| | Total | 237 | 61,421.62 | 261.84 | | | |

**Notes.**
d.f., degrees of freedom; MS, Microsatellite.

Areas with the highest suitability for the occurrence of *N. squamipes* in the present run almost continuously along the coast (Fig. 6A). The surface area suitable for *N. squamipes* in the present is 1,270,920.73 km$^2$ (considering the 10% threshold). The projection for the LGM shows a 62% increment (2,053,436.92 km$^2$), with the most suitable areas on
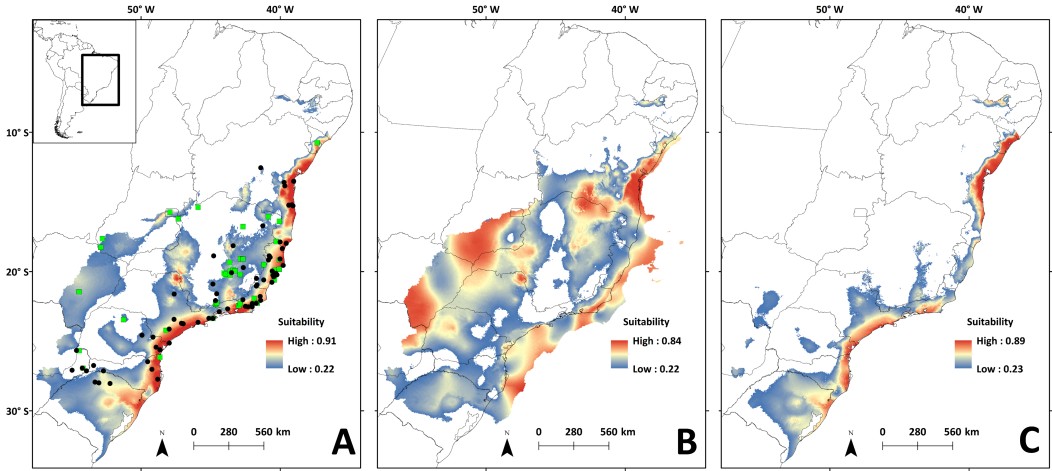

**Figure 6 Species distribution models for the river basins where *N. squamipes* occurs.** Dots represent the points used in building the model, and the square the points used in external validation. A: present; B: LGM, Last Glacial Maximum (22 ka); C: LIG, Last Interglacial (120–140 ka). ka, thousands of years ago.

the coast and inland near the headwaters of the São Francisco river, on the Espinhaço range, and along the Paraná river (Fig. 6B). The coast was displaced to the east during the LGM, exposing the continental shelf, which has areas of high environmental suitability for *N. squamipes*. Such areas run almost continuously from north to south, with a marked break at Serra da Bocaina, an area that shows lower suitability in the present as well (Fig. 6A). The LIG model indicated a sharp reduction in the suitable area for the species (620,808.98 km$^2$), a 51% decrease compared to the present, and a 70% decrease from the LGM. Highly suitable areas are restricted to the coast, with a break between Bahia and Rio de Janeiro (Fig. 6C). Therefore, we detected three major suitable areas in the LIG: two larger areas, one in the south and the other in northeast Brazil, and a third area around Rio de Janeiro in southeast Brazil. These three areas were also recovered in the LGM and in the present-day (Figs. 6A–6B), and they are similar to the geographical location of the three clades in the mitochondrial DNA analysis (Fig. 2).

## Landscape genetics

Cyt b and microsatellite flow paths indicated gene flow between distinct river basins, but D-loop paths pointed to more gene flow within river basin, regardless of temporal scale (Fig. 7). The microsatellite flow maps (Figs. 7C and 7F) is very similar to the Cyt b maps (Figs. 7A and 7D), and they are both distinct from the D-loop maps (Figs. 7B and 7E). The main difference between Cyt b and microsatellites is the presence of gene flow extending further south in the latter (Figs. 7C and 7F). The dispersal cost maps were created using only climatic variables, but the paths coincided with some main rivers in the Atlantic Forest. Four main paths among river basins were identified (Fig. 7): (1) along the coast between Paraíba do Sul and Jequitinhonha rivers (and Tietê for microsatellite data), connecting the East Atlantic and Southeast Atlantic basins (and South Atlantic basin for microssatellites); (2) along the Doce River and its tributaries, connecting the Southeast Atlantic basin to the

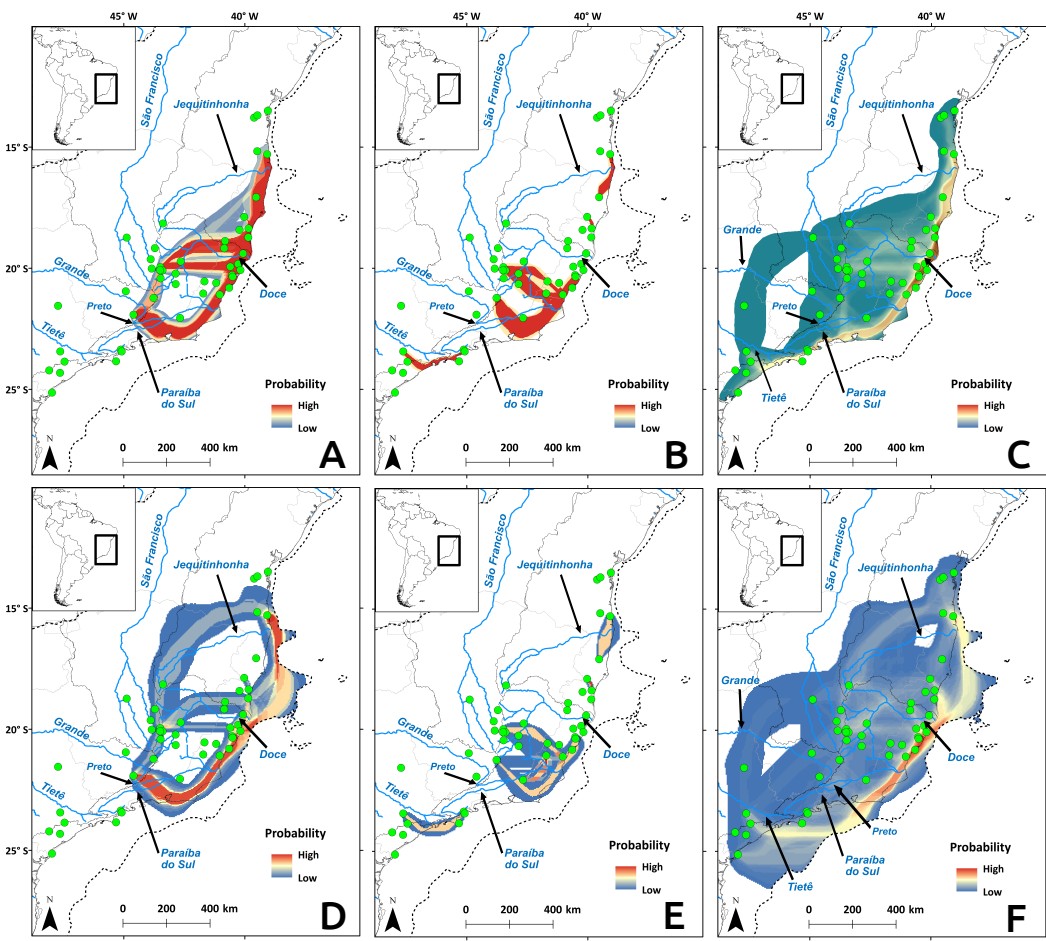

**Figure 7** **Probability of gene flow paths for *Nectomys squamipes*.** Connections among localities with shared Cyt b (A and D), and D-loop (B and E) haplotypes, and same allele size for all microsatellite loci (C and F). The gene flow paths were built using the SDM for the LGM as a friction layer (A–C), and the present-day model (D–F). Dotted line indicates the continental shelf exposed during the LGM, and submerged in the present-day.

others; (3) along the Tietê River and its tributaries, connecting the Paraná, South Atlantic and the Southeast Atlantic basins; and (4) along the Preto River connecting the Grande River sub-basin, which belongs to the Paraná River basin and Southeast Atlantic basin. Gene flow maps from the LGM showed differences in the use of the exposed continental shelf depending on the molecular marker: Cyt b (Fig. 7A) and microsatellites (Fig. 7C) showed that highest probabilities for gene flow paths are within the continental shelf, but not D-loop (Fig. 7B).

## DISCUSSION

### Genetic structure and historical connections among river basins
Our results showed a shallow genetic structure in *N. squamipes,* and mitochondrial DNA and microsatellite clusters are not strongly related to the five river basins. All molecular

markers showed a latitudinal genetic structure of the water rat, with three clades: north, central and south, with SDMs for different temporal scales recovering suitable areas in the same geographical location of these three clades.

The only results indicating strong genetic structure by river basin were $F_{ST}$ values for mitochondrial markers, which may be due to its matrilineal history and smaller effective population size when compared with microsatellite markers (*Avise, 2000*). Indeed, female *N. squamipes* have smaller home ranges and move less than males (*Lima, Pinho & Fernandez, 2016*), which may indicate that they are philopatric to natal sites. The small number of microsatellite loci used and their capability of detecting the true population structure is another plausible explanation for the discrepancy between this marker and mtDNA. Other recent studies using a similar number of microsatellite markers in other organisms (*Pérez-Alquicira et al., in press*; *Villamor, Costantini & Abbiati, in press*; *Zheng et al., in press*) also found significant genetic structure in some cases. Another study using the same microsatellite loci (*Maroja, Almeida & Seuánez, 2003*) found significant genetic differentiation in *N. squamipes* samples from a very small geographic area (maximum distance among sites = 6 km). We therefore believe that the number of microsatellites loci were adequate infer the genetic structure in our dataset.

The phylogeographic break at Serra da Bocaina is concordant with the boundaries of two freshwater ecoregions, Ribeira do Iguape and Paraíba do Sul, delimited by the Freshwater Ecoregions of the World (FEOW) system provided by *Abell et al. (2008)*. This is, however, the only break matching the FEOW, and the three clades do not follow the FEOW divisions. More samples, especially from the state of Rio de Janeiro, are needed to better delimit this phylogeographic break at Serra da Bocaina.

We considered the genetic structure of mitochondrial data is relatively shallow because there are: (i) few statistically supported clades; (ii) clades with short branches in the phylogenetic trees; and (iii) few isolated haplotypes in the haplotype networks. These same factors were found in the small semiaquatic rodent, *Arvicola sapidus* Miller (*Centeno-Cuadros, Delibes & Godoy, 2009*). In addition, the *N. squamipes* clades are separated by small genetic distances, which confirm low degree of differentiation (*Almeida et al., 2005*). Genetic studies on some small semiaquatic rodents, such as the muskrat *Ondatra zibethicus* (Linnaeus) (*Laurence, Smith & Schulte-Hostedde, 2013*) corroborated the shallow genetic structure and low divergence observed when the opposite was expected. Large semiaquatic mammals, such as beavers (*Castor fiber* Linnaeus) and otters [*Lutra lutra* (Linnaeus)], also display a shallow genetic structuring among populations in different river basins (*Durka et al., 2005*; *Quaglietta et al., 2013*). These cases may be explained by their greater dispersal ability as a function of body size and by the hunting pressure that led to a bottleneck effect (*Durka et al., 2005*; *Quaglietta et al., 2013*). Even for populations of the largest rodent, the capybaras (*Hydrochoerus hydrochaeris* (Linnaeus)), the importance of river basins is variable according to river system (*Byrne et al., 2015*). These results indicate that the dependency on rivers could favour the displacement of semiaquatic mammals and, as a consequence, gene flow, resulting in low divergence and shallow genetic structure. For instance, muskrats have an opportunistic behavior in the use of terrestrial landscape and population structure was not influenced by watershed network or landscape composition, and landscape features had

a limited effect on gene flow (*Laurence, Smith & Schulte-Hostedde, 2013*). Some specialist species may be considered generalist in their use of the landscape for dispersal, such as the southern water vole (*Centeno-Cuadros et al., 2011*), which uses several landscape types during dispersal, increasing gene flow. There is no landscape use data available for *N. squamipes* during dispersal but there are records that this species is capable of moving by land between forest fragments in the Atlantic forest during the dry season (*Pires et al., 2002*; *Lima, Pinho & Fernandez, 2016*), and the genetic structure we found suggests a more generalist behaviour during dispersal events, allowing individuals to move between basins.

There has been a recent population expansion in the Southeast Atlantic and Central clade/cluster according to the neutrality tests. Moreover, there are haplotypes in the Southeast Atlantic shared by other river basins, which indicates gene flow from adjacent basins is taking place now or in the recent past because the shared haplotypes concentrate at the borders. The microsatellite data, on the other hand, suggest stability rather than expansion in the Atlantic Southeast. This incongruence might be the result of the distinct temporal scales documented by these markers. Mitochondrial loci reflect older events than microsatellite loci (*Putman & Carbone, 2014*) and, in this context, the expansion detected by Cyt b and D-loop happened earlier than the recent populations stabilization documented by microsatellites. The same argument is valid for the differences between neutrality tests for the species as a whole, which indicated expansion of effective mitochondrial population size and stability based on microsatellite data. Moreover, SDMs showed that there is an increase of suitable areas in the Southeast Atlantic region from LIG to LGM, supporting the possibility of population expansion.

The gene flow paths among different basins occur at specific areas. The watersheds at the headwaters of rivers that delimit a basin may facilitate the connection among basins due to tectonic activity or erosion (*Ribeiro, 2006*). Important tectonic activity in the past 1.6 million years in the Atlantic Forest region may have caused reorganisation of river basins due to the capture of the headwaters of other rivers (*Saadi et al., 2002*). This phenomenon may affect both the migration of species among rivers and the current aquatic and semiaquatic organism communities. The shared fish fauna between river basins documented by several authors may indicate connection between them during the Quaternary (*Ribeiro, 2006*), which is the diversification period of *N. squamipes*. For example, *Pereira et al. (2013)* found phylogeographic structure in an Atlantic forest trahira *Hoplias malabaricus* (Bloch) nearly identical to presented here for *N. squamipes*. They explained this structure by recent dispersal events congruent with historical connections among river basins caused by sea level variations and stream piracy. The same factors may also explain the connections among river basins found here for *N. squamipes* populations (for details, see Fig. S3.5).

## Responses to past climate changes

Temperature is an important factor in the geographic distribution of *N. squamipes*, lower mean diurnal range of temperature (Bio 02) was the most important variable for constructing the distribution models. This variable is directly related to the biology of this water rat because it lives near water bodies where the temperature is more stable

during the day, and therefore coastal areas with narrow temperature range are highly suitable in the SDM. Population fluctuations over the past 120,000 years followed the temperature variations documented by the stable isotope oxygen-18 ($\delta^{18}O$) (*Cohen, 2012*), with an expansion in the LGM, but the confidence interval is quite wide. We also found evidence of population expansion in the neutrality tests, and the expansion of suitable areas through time is consistent with an expansion in the LGM when the climate was cooler. The evidence of population and range expansion in the LGM and contraction of suitable areas with increasing temperature both in the present and in the LIG contradict the forest refuge hypothesis to explain the biogeographic history of this species. The phylogeography of *N. squamipes* documented here fits the Atlantis forest hypothesis, which predicts the expansion of Atlantic forest onto the continental shelf during the LGM. This is in accordance with our estimate of population sizes, our species distribution model for the LGM, as well as coastal gene flow pathways displaced to the now submerged continental shelf. D-loop was the only genetic marker that did not show gene flow paths on the continental shelf during the LGM. The rate of evolution in D-loop sequences is intermediate between the slower Cyt b and the faster microsatellites, therefore D-loop flow paths probably result from distinct events that took place in an intermediate time, such as an interglacial period, when dispersal was restricted.

*Nectomys squamipes* populations may have been favoured during glacial periods despite possible decrease in plant cover and increase in rainfall seasonality (*Behling, 2002*; *Carnaval & Moritz, 2008*). This scenario may have reduced the habitat of other semiaquatic mammal because of lower river level, while favouring *N. squamipes*, a more generalist species that prefers small marshy streams over larger rivers with faster-running water (*Lima, Pinho & Fernandez, 2016*). Another factor that may have contributed to an increase in the distribution of *N. squamipes* during the LGM is the decrease in ocean levels, which were on average 68 m lower than their current level (*Lambeck, Esat & Potter, 2002*; *Rabineau et al., 2006*). This increased not only the land surface, but also extended the course of rivers flowing into the Atlantic Ocean (*Weitzman, Menezes & Weitzman, 1988*; *Clapperton, 1993*; *Thomaz et al., 2015*), allowing connections that are now hidden by the ocean, and favouring dispersal and gene flow during this glacial period. When the sea rose to its current level, coastal populations were isolated in the higher areas that became oceanic islands, which is in agreement with the current genetic structure of island populations of *N. squamipes* (*Almeida et al., 2005*). Therefore, riparian forests may have functioned as corridors among forest fragments because they have the same resource availability (*Naxara, 2008*). Pleistocene palaeodrainages were crucial in structuring genetic diversity in fish (*Thomaz et al., 2015*), and these palaeodrainages may have played an important role shaping the genetic structure of *N. squamipes* as well.

Dating revealed a Pleistocene origin for the Cyt b haplotypes of *N. squamipes* and for the three main clades. This result indicates that climate oscillations in the period may have been important both for the speciation of the genus and for the genetic structuring of *N. squamipes*. The three main geographic regions resulting from modelling the LIG are similar to the clades found in the phylogenetic trees, indicating correspondence between the modelled ecology data and the genetic data, as described in other studies (*Igea et al.,*

*2013*). Taking into account that there were several glacial periods during the Pleistocene, with interglacial periods in between, the *N. squamipes* populations may have undergone recurring processes of population reduction and expansion (*Almeida et al., 2000a*).

## CONCLUSIONS

Our results showed a shallow genetic structure in the water rat, slightly influenced by river basins. The pattern of genetic diversity is best explained by latitude with three main clades structured from north to south. There is evidence of recent population and suitable area expansions during glacial period for *N. squamipes*. The presence of suitable areas and gene flow paths through the continental shelf, and population expansions during the glacial period, give support to the Atlantis forest hypothesis. Rivers and historical connections between rivers allowed *N. squamipes* to disperse farther across and inside basins, leading to a shallow genetic structure.

## ACKNOWLEDGEMENTS

CR Bonvicino, LP Costa, MA Sábato, M Lara, M Passamani, R Moura, V Fagundes and the collections of the Museu de Biologia Professor Mello Leitão (MBML), Museu de Ciências Naturais da PUC-Minas (MCN-M) and Universidade Federal do Espírito Santo (UFES-CTA), which all donated tissue samples for this study. KPMB Ferraz, MC Ribeiro, FC Barreto and JP Hoppe provided great help with the distribution modelling, TE Simon (in memoriam) and JF Justino with microsatellite analysis, H Seibel Jr. with Python scripts, KA Boher with BAPS analysis. AB Santos, V Fagundes, A Percequillo, AT Thomaz, AC Loss, and anonymous reviewers made suggestions that improved the quality of earlier drafts of this manuscript.

### Funding

Fundação de Amparo à Pesquisa e Inovação do Espírito Santo (FAPES) and Conselho Nacional de Desenvolvimento Científico e Tecnológico (CNPq) financed this project (grant # 385/2011 to Yuri Luiz Reis Leite). Jeronymo Dalapicolla received a scholarship from the Coordenação de Aperfeiçoamento de Pessoal de Nível Superior (CAPES) and Yuri Luiz Reis Leite received a scholarship from CNPq. Currently, Jeronymo Dalapicolla receives a scholarship from the Fundação de Amparo à Pesquisa do Estado de São Paulo (FAPESP; grant #2015/02853-6, and #2016/24464-4). The funders had no role in study design, data collection and analysis, decision to publish, or preparation of the manuscript.

### Grant Disclosures

The following grant information was disclosed by the authors:
Fundação de Amparo à Pesquisa e Inovação do Espírito Santo (FAPES).
Conselho Nacional de Desenvolvimento Científico e Tecnológico (CNPq): # 385/2011.
Coordenação de Aperfeiçoamento de Pessoal de Nível Superior (CAPES).

CNPq.

Fundação de Amparo à Pesquisa do Estado de São Paulo (FAPESP): #2015/02853-6, # 2016/24464-4.

## Competing Interests

The authors declare there are no competing interests.

## Author Contributions

- Jeronymo Dalapicolla conceived and designed the experiments, performed the experiments, analyzed the data, prepared figures and/or tables, authored or reviewed drafts of the paper, approved the final draft.
- Yuri Luiz Reis Leite conceived and designed the experiments, contributed reagents/materials/analysis tools, prepared figures and/or tables, authored or reviewed drafts of the paper, approved the final draft.

## DNA Deposition

The following information was supplied regarding the deposition of DNA sequences:

The mitochondrial markers sequences described here are accessible via GenBank accession numbers KY498357–KY498476.

## Data Availability

The raw microsatellite data are available in the Supplemental Files.

## Supplemental Information

Supplemental information for this article can be found online at http://dx.doi.org/10.7717/peerj.5333#supplemental-information.

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
