# Peer review of "Historical connections among river basins and climatic changes explain the biogeographic history of a water rat"

_PeerJ, doi:10.7717/peerj.5333_

## Round 0.1 · original submission · Major Revisions

All of the requests for clarification, suggestions for improvement and new analyses by the reviewers are quite pertinent. I am particularly worried about the pseudogene issue, since their occurrence may not be random and bias the results: did failing samples cluster by location/watershed?

I also think that the BSP results should not be overinterpreted: in figure 6, a flat horizontal line could fit within the confidence interval, so a demographic history with no changes in effective population size is also supported by your results. By the way, that figure lacks a y-axis scale.

l. 77: borader -> broader

l.391-394 and 417-421 are practically identical. Please paraphrase and abridge the second instance of this text.

Reviewer 1 ·

Basic reporting

no comment

Experimental design

no comment

Validity of the findings

no comment

Additional comments

This work addresses the phylogeographic structure and biogeographic history of the water rat Nectomys squamipes from 50 localities of different river basins, using two mitochondrial genes and six microsatellite loci. Various types of analyses are applied to evaluate evolutionary hypotheses that explain the biogeographic history of the species and test if the genetic structure is influenced by hydrography. This work may become a relevant contribution of an interesting species but I think there are some minor issues as well as important inconsistencies that should be clarified.

Line 89: You may briefly explain how the sampling was performed or you may give the sources from which samples were obtained, for example, in Appendix S1.

Lines 98-99. I do not understand how resequencing can help to detect pseudogenes. Please explain better the measures taken to avoid sequencing pseudogenes, as this is an important issue.

106-107. The thresholds for bootstrap values and posterior probabilities make no much sense here. Explain the values obtained in results.

136-137. Give references for the microsatellites if they were not developed in this work.

150-151. Why using a priori classification by river basins in Structure? If there is no a justification, the results may be more meaningful if you use Structure without any a priori classification.

176-177. It should be obvious that you verified locations with any mapping tool like Google Maps or Google Earth, so you do not need to mention both tools. What it is far from obvious is how you used SpeciesLink and what is the meaning of geoLoc, infoXY, etc. Please explain better if this is relevant.

223-225. The fact that pseudogenes were found for this species reinforces the need to better explain how their sequencing was avoided here.

230 and Figure 2. Several phylogenetic methods are explained in Method. Which specific tree is shown in Figure 2? In any case, this tree does not show branch lengths, only topology. It would be convenient to choose a representation in which branch lengths are shown. It would also be desirable to comment on the differences observed between trees reconstructed with different methods. Same for Figure S3.1.

231-233. You describe the tree with D-loop but then you refer to the tree figure with concatenated sequences. Please clarify.

233-235. It would be easy to test if the outgroups affect the topologies by removing the outgroups from the cytb alignment and comparing the trees with and without outgroups.

250-251. The results obtained microsatellite genetic structure should be given more relevance. They deserve at least being in their own paragraph and a more thorough explanation since microsatellites can reveal more accurately the population structure without the problems of the mitochondrial markers (as the authors comment in Discussion). Take also into account the possibility of using a model without a priori classification, as suggested in a previous point.

262-267. It would be interesting to compare the results of differentiation from mitochondrial data with those inferred from microsatellites using similar tests or, better, Structure with a proper model.

271-275. Since you are not using a coalescent framework, these dates do not reflect the common ancestor of all Nectomys and, particularly, of N. squamipes. These dates only represent the common ancestor of the haplotypes. Care should be taken when using these dates for biogeographical inferences, and possible caveats with these dates should be mentioned.

313-316. In these two consecutive phrases you say that results from mitochondrial markers are "expected" and then that they are "striking". Please clarify the idea.

317-321. I do not find it clear if your microsatellite data is robust or not from this paragraph. Please use additional references (other species, reviews, etc.) to support your data. Otherwise, if the microsatellite results are too weak, it may be better not to discuss them further.

330. Clades with short branches in the phylogenetic trees can only be appreciated in a tree figure with branches (see previous point).

357-365. Again, microsatellite results seem not to fit with the other data but I am not sure if this is simply due to different temporal scales, as the authors argue. As suggested in previous points, the way you pre-classified data by basin may complicate congruence among results.

409-410. The Atlantis forest hypothesis seems to be well supported for this species with the mitochondrial and SDM results. However, I do not understand how nuclear loci (microsatellites) can support the Atlantis forest hypothesis when other paragraphs explain the incongruence between mitochondrial and nuclear data. Unfortunately, the referred criticism that the hypothesis is only based on mitochondrial data may remain. As already stated, a greater effort to analyze microsatellite data should be made and, if the results are relevant, explain better how they support or refute different hypotheses.

426-427. As already stated, your dating analysis cannot be used to date the origin of this and other species.

437. Again, I don't think you can conclude that there is congruence between mitochondrial and nuclear data. This contradicts most of your discussion.

Figure 1: Check the map scale, which seems to be wrong.

Figure 4: I suppose you mean "bar graphs".

Figure 8: Maps should show the same extent, if possible, for easier interpretation.

Reviewer 2 ·

Basic reporting

No comment

Experimental design

It is not possible to arrive at one of the two main conclusions of the manuscript based on the sampling design used

Validity of the findings

No comment

Additional comments

Review: Historical connections among river basins and climatic changes explain the biogeographic history of a water rat (#25252)


I have read the above mentioned manuscript with attention. This manuscript investigates the phylogeographic structure of N. squamipes using mitochondrial and nuclear markers in order to evaluate evolutionary hypotheses to explain the biogeographic history of the water rat and test if the genetic structure is strongly influenced by hydrography. This is a very interesting manuscript, well written and structured. The introduction clearly presents the background in the subject of study and the results are presented in a clear manner. However, sampling design and the interpretation of some results present certain flaws.
I think that there is scope for this study to yield a paper of interest to a large audience. Unfortunately, I have identified some issues (listed below), which, I believe, need to be dealt with prior to publication. I hope that the authors find my comments below useful.

General comments:

One of the main conclusions of the manuscript is that rivers allow N. squamipes to disperse farther across and inside basins, leading to genetic homogeneity and a shallow genetic structure slightly influenced by river basins. I believe that this conclusion should be reviewed carefully. Firstly, the sampling performed is not representative of all the analyzed basins, since basins such as SeA or EA present between 35 and 60 individuals sampled (Table S 2.4) and basins such as Paraná, SA and SF present only 6 to 13 individuals sampled. This difference in the number of individuals analyzed per basin could be the cause of the lack of genetic differentiation between them only as consequence of a bias in the sampling.
Additionally, AMOVA analyses performed with the mitochondrial control region, a more appropriate marker than cytochrome b for studying genetic structure, showed significant differences between basins (Fct = 0.25, P <0.01, Table 4) and between pairs of basins (Table 3), contradicting the idea of genetic homogeneity between river basins. This result should not be underestimated by the authors, given the importance of maternal lineages in the conformation of genetic structure.
With the aim of solving the bias in basins sample size and trying to make a better use of sequence data with more innovative methodologies, I encourage the authors to add the Bayesian analysis using BAPS or Geneland software in order to estimate the number of genetic cluster present in the study area and then to test for genetic differences between clusters independently of river basin.

Minor comments

Lines 65-66: I think this sentence should be deleted as it is confusing. It is very difficult to imagine that rivers could lead to isolation in this species. Or do the authors refer to high flow rivers that could act as barriers to dispersion? I think that in the case mentioned by the authors, the specialization would be the cause of the isolation and not the rivers.

Line 89: How were the samples taken? Were live individuals captured and sacrificed for liver samples? Or were dead individuals collected? In the case of using live individuals and sacrificing them, had the authors the necessary permits? Please explain in more detail.

Lines 95=96: Only cyt b electropherograms were carefully examined? What about d-loop ones?

Line 103: Which models were selected?

Lines 133-135: Why the BSP was conducted with cyt b sequences only?

Line 143: Why were k values chosen between 1 and 6? How many genetic clusters were expected?

Line 158: Why was MaxEnt chosen among all the available species distribution models? Did you try some other model?

Line 226: If 152 of the 154 individuals sampled were genotyped. Why only 42 of the 50 sampling locations were represented?

Lines 243-244: “Genetic divergences were small within and among Nectomys species, among the N. squamipes clades, and among river basins”. These results must be explained in greater detail. Only the results of intraspecific divergence are displayed

Lines 246-247: I do not agree that an interclade divergence of 4.0-4.6% for the control region is small. This differentiation corresponds in many cases to that observed between different mammal species.

Lines 262-265: “We found a very large (FST>0.25; Hartl & Clark, 1997) mitochondrial (cyt b and D-loop) differentiation among river basins (Table 3). The largest FST values were found between the Southeast Atlantic and South Atlantic basins (cyt b: 0.74; D-loop: 0.64), indicating little divergence within basins and large divergence between basins”. These results also contradict the idea of genetic homogeneity between basins.

Lines 267-270: It is striking that the AMOVA analysis carried out using cyt b sequences showed non-significant differences and all the paired-Fst comparisons between basins are significant. Please explain.

Lines 315-316: This is not striking at all. This type of patterns is very common in different species of mammals in which males disperse and females are philopatric.

---

## Round 0.2 · Minor Revisions

Please address the remaining issues raised by reviewer 1

Reviewer 1 ·

Basic reporting

No comment.

Experimental design

No comment.

Validity of the findings

No comment.

Additional comments

The authors have addressed the most important concerns and the manuscript has improved with respect to the previous version. However, there are still a few minor issues that the authors should clarify:

Abstract, line 15: "that shows genetic homogeneity across space". It is strange to say in the first phrase of the abstract that the species is homogeneous when the introduction makes clear that this is a controversial issue, with some authors having found a relevant genetic structure (lines 75-77). I also think that "genetic homogeneity" is not properly used in line 30 of the abstract and in several other parts of the manuscript. The authors are in the safe side saying that they found a "shallow genetic structure" because they talk about three mitochondrial clades, etc. but this is not "genetic homogeneity".

In general, the authors are advocating too strongly for the "genetic homogeneity" explanation and even give some examples that do not really support it. For example, Galemys pyrenaicus (line 379) shows a particularly strong genetic structure both at large and small scales, so this is not a good example neither for "shallow genetic structure" nor for "genetic homogeneity" and can be removed from here. In general, it will be difficult to find examples of mammals showing genetic homogeneity, but clearly, there are many with shallow genetic structure. In addition, in this part of the discussion the authors sometimes seem to equate small genetic divergence with low genetic structure but these are two different concepts and the authors should be more careful with them.

The issue about pseudogenes is better explained now, at least for cytochrome b. However, given the big problems found with this gene, one could think that they may also be present in the D-loop, at least at some frequency, but nothing is commented on this. The authors could comment in Results if some pseudogene was found for the D-loop or if they were only present in the cytb gene.

In the justification for using the locality prior in Structure in Methods (line 160), you should state that these recommendations are for situations where the amount of available data is very limited (few markers or few individuals), as indicate in the response to reviewers.

Reviewer 2 ·

Basic reporting

I would like to congratulate the authors on their hard work, for addressing the comments from a previous review of the manuscript. Authors have done a great job. I consider that there has been a substantial improvement since the first version. I enjoyed reading the manuscript and recommend considering it to be published in PeerJ.

Experimental design

No comment

Validity of the findings

No comment

---

## Round 0.3 · accepted · Accept

Thank you for this last round of edits

#